# Biology, Systematics, Microbiome, Pathogen Transmission and Control of Chiggers (Acari: Trombiculidae, Leeuwenhoekiidae) with Emphasis on the United States

**DOI:** 10.3390/ijerph192215147

**Published:** 2022-11-17

**Authors:** Kaiying Chen, R. Michael Roe, Loganathan Ponnusamy

**Affiliations:** 1Department of Entomology and Plant Pathology, College of Agriculture and Life Sciences, North Carolina State University, Raleigh, NC 27695, USA; 2Comparative Medicine Institute, North Carolina State University, Raleigh, NC 27695, USA

**Keywords:** chiggers, biology, systematics, microbiome, pathogen transmission, control

## Abstract

Chiggers are the larval stage of Trombiculidae and Leeuwenhoekiidae mites of medical and veterinary importance. Some species in the genus *Leptotrombidium* and *Herpetacarus* vector *Orientia* species, the bacteria that causes scrub typhus disease in humans. Scrub typhus is a life-threatening, febrile disease. Chigger bites can also cause dermatitis. There were 248 chigger species reported from the US from almost every state. However, there are large gaps in our knowledge of the life history of other stages of development. North American wide morphological keys are needed for better species identification, and molecular sequence data for identification are minimal and not clearly matched with morphological data. The role of chiggers in disease transmission in the US is especially understudied, and the role of endosymbionts in *Orientia* infection are suggested in the scientific literature but not confirmed. The most common chiggers in the eastern United States were identified as *Eutrombicula alfreddugesi* but were likely misidentified and should be replaced with *Eutrombicula cinnabaris*. Scrub typhus was originally believed to be limited to the Tsutsugamushi Triangle and the chigger genus, *Leptotrombidium*, but there is increasing evidence this is not the case. The potential of *Orientia* species establishing in the US is high. In addition, several other recognized pathogens to infect humans, namely Hantavirus, *Bartonella*, *Borrelia*, and *Rickettsia*, were also detected in chiggers. The role that chiggers play in these disease transmissions in the US needs further investigation. It is possible some of the tick-borne diseases and red meat allergies are caused by chiggers.

## 1. Introduction

Chiggers are the larval stage of the families Trombiculidae and Leeuwenhoekiidae mites (Phylum Arthropoda, Class Arachnida, Order Trombidiformes, Superfamily Trombidioidea, and Family Trombiculidae, Ewing, 1929 and Family Leeuwenhoekiidae, Womersley, 1945. They are also called red bugs, harvest mites, berry bugs, scrub-itch mites in English-speaking countries, and sand mites in China. Chiggers are of medical and veterinary importance. Some species in the genus *Leptotrombidium* and *Herpetacarus* [1,2] are known to vector a Gram-negative bacterium, *Orientia* species (formerly in the genus *Rickettsia*), causing the disease, scrub typhus, in humans. Scrub typhus is a life-threatening, febrile disease. During the Second World War, a large number of scrub typhus patients were found in the Asia-Pacific region [3], which resulted in greater attention and research on all aspects of the disease. Humans are only an occasional host for chiggers. Chiggers parasitize many different vertebrates, including reptiles, amphibians, birds, and mammals, and were reported to be significant pests for free-range turkeys in the US [4]. Their bite frequently results in dermatitis (trombiculosis or trombiculiasis).

Publications of chigger studies in the US were searched on the Web of Science on 3 March 2022 (Appendix A), showing 115 papers of which 83 were directly on chiggers. The number of publications on chiggers in the US is shown in Figure 1. Publications were summarized and shown in Appendix A. The publications mainly focused on taxonomy (identification methods), ecology, and reports of the discovery of chiggers on various hosts in the US. Chigger research in the US started in the 1910s. The number of publications gradually increased from 1920 to 1979 followed by a drop in the 1980s. From 1990 till now, the number of publications per decade on this topic was 13–16. However, these data do not include all of the existing published papers, especially the older papers. For example, as stated in the world checklist of Trombiculidae and Leeuwenhoekiidae from Nielsen et al. [5], H.E. Ewing described 64 new chigger species from 1913 to 1950. Nielsen et al. listed 23 Ewing’s publications and five were marked as cited by other researchers. Based on our search results, there were only 10 papers published by Ewing that were available on the Web of Science, but additional papers were available at our institutional library.

It is challenging to find papers about chigger studies in the US published 50–90 years ago. Even for the publications that can be found on the Web of Science, some are not easily accessible for review. Publications after the 1990s were better documented. It is clear that chiggers in the US are greatly understudied in general and especially as compared to other vector important arthropods like mosquitoes, ticks, lice, fleas, and filth flies. The goal of this paper was to review the biology, systematics, microbiome, pathogen transmission and control of chiggers with emphasis when possible on what we know about them in the US.

## 2. Chigger Biology

### 2.1. Life Cycle

The life cycle of chigger mites is comprised of seven stages: egg, quiescent prelarva (also called deutovum), parasitic larva, quiescent protonymph, soil free-living deutonymph, quiescent tritonymph and soil free-living adult (Figure 2). The two active post larval stages, deutonymph and adult, have eight legs, live in soil, and are predators feeding on arthropod eggs and other life stages of small, soil-dwelling insects and other arthropods. The parasitic larval stage, so-called chiggers, have six legs. Unengorged chiggers are about 0.2 mm long and deep-red in color, while engorged chiggers are about 0.4 mm long and pale yellow [6,7].

Chiggers do not burrow into skin and feed on blood as many may think. Instead, chiggers feed on saliva digested lymph and skin cells. They insert their chelicerae into the skin and secrete saliva containing proteolytic enzymes into the skin. The epidermal and cutaneous host tissue is liquefied, and the liquid is consumed. As the feeding progresses, a hard tube-like structure, called a stylostome, is formed by the salivary secretions and the host immune system responds to the foreign enzymes in the chigger saliva [8]. The stylostome functions as a straw, allowing the chiggers to consume the exo-digested skin. The saliva secretion and stylostome formation from feeding cause an inflammatory dermal reaction in the host [9], which is called trombiculiasis. Chiggers will attach to a host and feed for 1–12 d if not scratched off [10,11,12], the duration variable depending on the host species. After engorgement, chiggers drop naturally from the host to the soil and become quiescent.

Overall, the available data for the developmental time for each stage both in the field and lab are limited. In optimal lab conditions (25–30 °C, 100% relative humidity), a generation of *L. deliense* is 59–135 days, with an 89 days average [13]. The development time of a complete generation for boreal species can take anywhere from 150 to 400 days [14]. In the field, local climate conditions will obviously affect the developmental time since chiggers are ectotherms. Wharton and Fuller [15] reported *E. alfreddugesi* produced one to two generations a year in Ohio, three generations in North Carolina and developed year-round in Florida, US. Direct copulation between males and females has never been observed. The adult female is inseminated by making contact with a spermatophore deposited by males in the habitat that they share [16]. The male spermatophore is made up of a flexible stalk (about 5 µm thick and 45 µm tall) that is linked to the substrate and bears a sperm sac at the distal end [11,16]. When the female encounters the spermatophore, she will walk over the spermatophore with her genital plates fully opened and pick up the sperm sac. Lipovsky et al. [16] observed that chigger females could acquire multiple spermatophores. Females normally undergo one to three oviposition cycles with gaps between them throughout their lives [14].

### 2.2. Hosts and Habitat

The host spectrum of chiggers is broad. For example, *E. alfreddugesi*, the most widespread chigger species in the US, were found to parasitize reptiles, amphibians, birds and mammals. Walters et al. [17] listed that 145 species of reptiles and amphibians, 129 species of birds, and 225 species of mammals in the US were reported to be parasitized by chiggers. Some chigger species or genera were only reported parasitizing a particular host group. For example, almost all *Neoschoengastia americana* parasitize birds [17] and this species was reported to be a significant pest of turkeys in the US [4]. Chiggers in the genus *Hannemania* primarily fed on amphibians [17]. However, the preference of chiggers to a specific host species is low. Chiggers are habitat-specific rather than host-specific. Chiggers are known to have preferences for a certain local habitat, where they attack and feed on all or the majority of the vertebrate species in the same location [14,17,18,19]. The low host specificity of the chigger and broad host range may enhance their chances of contacting humans, as well as the spread of scrub typhus across diverse hosts. Scrub typhus so far has not been reported in the US (discussed in more detail later).

Chiggers prefer moist areas with overgrown vegetation [15], like abandoned plantations, tall grass, and river banks. Chiggers are found to be most abundant in early summer in temperate regions when the vegetation is heaviest. Clopton and Gold [20] reported the abundance of *E. alfreddugesi* peaked in late June to early July. In the summer, adults were most common near the soil surface, and in the fall and winter, they were found deeper in the soil [15]. Shatrov and Kudryashova [14] reported the eggs of some boreal chigger species can diapause for up to 400 d. This suggests they have temperature preferences likely related to their life cycle and survival when environmental conditions are not favorable for development.

### 2.3. Collecting Methods

#### 2.3.1. Trapping Animals

This method of collecting requires approval from Institutional Animal Care and Use Committees (IACUCs). Rodents can be live-trapped with Sherman traps (Figure 3A) placed on the ground overnight at any place where chiggers and potential hosts are expected to be found [21]. Chiggers then can be collected from rodents the next day. Sherman traps are usually baited with bird seed [22] or a peanut butter and oats mixture [23]. Ketamine hydrochloride and xylazine sulphate (1:10) are given intramuscularly to anesthetize the captured rodents [22]. Captured rodents are examined thoroughly for the presence of chiggers with particular attention to the ears, groin, foot, belly and area surrounding the genitals [23]. Removed chiggers are preserved in a vial containing 70% ethanol [22]. Similar approaches may be used for other types of animals like birds, lizards, or larger mammals. One advantage of this approach, the rodents are traveling throughout their natural geographical range foraging for food, obtaining nesting materials and mating and have a greater chance of acquiring chiggers from their habitat than other collection methods (discussed later). Additionally, the chiggers are easy to collect from the animal after trapping, since they are attached to specific locations, e.g., in the ears of mice. This approach is only effective for obtaining feeding chiggers. Since they are feeding on animals that could be infected with a human pathogen, for example, their microbiomes might reflect that of the host and not necessarily that of a free-living chigger.

#### 2.3.2. Plate Method

Unengorged chiggers can be attracted to foreign objects of almost any kind [24], for example, a porcelain plate, plastic disk, or oilcloth [21,24]. Black plates (Figure 3B) are attractive to unfed chiggers when heated by the sun, and the black color makes it easy to see the reddish color of chiggers on the plate, especially when they are crawling. Thus, black plates were commonly used to collect unengorged, free-living chiggers in rodent runways and near nests [25,26,27,28]. A set of black plates (10 cm by 10 cm) are laid at a possible breeding site at intervals. Each black plate is observed for any movement after a certain amount of time (e.g., 10 min). Chiggers can be collected with an ethanol-dipped pipette or small camel hair or equivalent brush and then transfer to a tube containing 70% ethanol. A hand lens helps ensure a successful transfer. Another collecting method is using an inverted white tray placed on the ground in a chigger infested area overnight; when the tray is righted the next day, free-living chiggers can be collected from what was the undersurface of the tray [21,24].

The advantages of the plate method, free-living chiggers are collected alive and in good viable condition, in case the researcher wanted to develop a laboratory colony or study the live biology, physiology or microbiome of the chiggers. The challenge with this method is finding an exact location where the chiggers are found. This is typically a random approach within a potential geographical location where chiggers are expected. Sometimes, picking a place to collect can be enhanced by incidental reports of chigger problems, frequently provided by park rangers or comments from visitors to a particular area.

#### 2.3.3. Berlese Funnel

A Berlese funnel (Figure 3C) can be used to collect chigger mites from soil simply by placing the litter sample into the inverted funnel. The mites will crawl downward from the light shining into the top, large opening of the funnel, as the soil dries and they eventually fall into a bottom collection chamber filled with ethanol (to kill and preserve the mites) positioned below the end of the inverted funnel [18]. The challenge with this approach would be the need to collect enough soil samples to find the mites and have enough Berlese funnels to process the soil samples, since the free-living chiggers are not typically widely and evenly distributed in the landscape. Additionally, other soil animals will drop into the ethanol and will need to be separated from the free-living chiggers. If there is an interest in the external microbiome of the chigger, there would be cross contamination between animals. The chiggers collected by this method are killed by the ethanol.

#### 2.3.4. Light Trapping

Johns [29] developed a light trap to collect unengorged chiggers in vegetation using the natural tendency for the unengorged chiggers to be positively phototropic with a preference to climb higher. The trap was made of a piece of 60.96 cm by 60.96 cm waterproof and lightproof paper with a 7.62 cm by 7.62 cm hole cut in the middle. A celluloid window measuring 8.26 cm by 8.26 cm was placed over the hole and attached to the paper with adhesive tape to make a central window frame about 6.35 cm by 6.35 cm (Figure 3D). The trap is laid in the desired location with the celluloid window in contact with the soil surface or the vegetation so the chiggers could climb from the dark field to the celluloid window (Figure 3E). The edges of the trap are secured by bricks, stones or wood pieces to hold the sheet to the ground and assure the only light entering the trap is was from the top celluloid window. When collecting the chiggers, the trap is turned over, and the chigger transferred by a wet brush onto a moistened filter paper in a collecting jar to extend the life of the chigger. This approach to keep the free living chiggers alive could also be used for the plate method of collecting (discussed earlier).

### 2.4. Chigger Lab Rearing

A chigger colony can be started from field-collected engorged larvae taken from trapped hosts. The larvae are allowed to detach from the host naturally to avoid killing them [30]. This will require a different IUCUC protocol for holding the wild-caught host longer than typically needed for collecting chiggers from a trapped host which are released the same day. Collecting chiggers for starting a lab colony is more challenging because the wild animals have to be maintained in environmental conditions to preserve the health and well-being of the host. For biosecurity reasons, wild hosts cannot be held in a rearing facility with other research animals because of the threat of introducing a disease.

Chiggers throughout their life cycle are maintained in 3.5 cm diameter plastic containers, 1/3 filled at the bottom with hardened plaster of Paris: charcoal (9:1), saturated with water [31]. Michener [32] also reported using fruit jars with the inner surface coated with 1/8 to 1/4 inch thickness of plaster of Paris to rear *Eutrombicula batatas* in the lab. Eggs of collembola, e.g., *Sinella curviseta* [33] or of mosquitoes, e.g., *Culex pipiens* [18], are provided ad libitum to nymphs and adults as food. Eggs before hatching are separated from the other stages by flooding the container; the eggs remain at the bottom [30]. The unfed chigger larvae are placed on a laboratory mouse in a tight-fitting wire-mesh cage where the activity of the mouse is restrained. After the chiggers have attached to the mouse for feeding, they (mouse and chiggers) are transferred to a loose-fitting cage [30]. Michener [32] also reported to use domestic chickens to be used as host of lab rearing chiggers. A tray (slightly bigger than the cage) filled with water and the cage are held above the surface of water and used to catch engorged larvae when they drop from the mouse [30]. For chiggers that do not parasitize mammals, snakes and turtles can be used to feed larvae [34]. The temperature for rearing *L. deliense* and *L. fletcheri* is 27 ± 3 °C [31]. The optimal relative humidity is 80–100% [14,21]. In addition, Audy and Nadchatram [35] reported method of rearing chigger larvae collected from the field in glass tube fitted with wet strip of paper. This method was for only taxonomic study purpose.

One challenge of starting a colony from the field is the potential for the engorged chiggers to be infected with human pathogens, especially if the chiggers were collected from known endemic areas of disease. Additionally, there are areas where the status of chigger pathogens have not been investigated nor the transmission potential. These unknown concerns can complicate starting a laboratory colony, since BSL level III containment that involves the use of vertebrate animals would be needed if human pathogens are involved or potentially involved as part of the rearing effort.

## 3. Systematics Status

### 3.1. Taxonomy

Three classification systems for chiggers have been suggested: (*i*) all chiggers are in the family Trombiculidae Ewing, 1929 with three subfamilies, Trombiculinae, Ewing, 1929, Leeuwenhoekiinae, Womersley, 1944 and Apoloniinae Wharton, 1947 [36,37,38,39]; (*ii*) all chiggers are in two families, Trombiculidae and Leeuwenhoekiidae Womersley, 1945 [5,17,40]; and (*iii*) all chiggers are in in three families, Trombiculidae, Leeuwenhoekiidae and Walchiidae [41,42,43]. There is disagreement on whether the taxa should be included in the family Trombiculidae [44]. In this review, we are using the classification that chiggers are the larval mites in the families Trombiculidae and Leeuwenhoekiidae.

Walters et al. [17] summarized all publications from 1923 to 2009 that reported chiggers in North America, the chigger species reported in each publication, the hosts that the chiggers were found, and the US state or Canadian province where they were associated. The list contained 42 species in Leeuwenhoekiidae and 205 species in Trombiculidae in the US [17]. In 2015, Crossley and Clement [45] reported a new species, *Hoffmanniella solickiana*, from the Rafinesque’s Big-Eared Bat in Georgia, US, making a total of 248 chigger species from the US, containing 42 species in Leeuwenhoekiidae and 206 species in Trombiculidae. The chigger species richness by state in the US is diverse (Figure 4).

Chiggers are reported in every state except Hawaii in the United States. The states that reported more than 30 chigger species were California (97 species), Texas (74), Utah (52), Kansas (47), Colorado (30) and Nevada (30). The chigger species were more diverse in the Southwest. Some species in the genus *Leptotrombidium* were found in the US, i.e., *Leptotrombidium myotis*, *L. panamensis*, and *L. peromysci*. The genus *Leptotrombidium* is known as the vector of *O. tsutsugamushi*, the causative agent of scrub typhus.

Based on the list from Walters et al. [17], *E. alfreddugesi* (Oudemans 1910) was the most widely spread chigger in the US, found in 22 states from 179 host species, including reptiles, amphibians, birds and mammals. However, Bennett et al. [46] argued that the most common chigger in the eastern US, *E. alfreddugesi*, was possibly misidentified and should be replaced with *E. cinnabaris*. Loomis and Wrenn [47] investigated the systematics of the genus *Eutrombicula* and found that this genus in the Eastern US needed revision since some of the records were misidentified. This species was named by Oudemans as *Microthrombidium alfreddugesi* in 1910 based on four specimens collected from Temascaltepec, close to Tejupilco, Mexico, which was the type species for the genus *Eutrombicula*, Ewing, 1938 [48]. Fuller [49] revisited the two representative specimens of Oudemans’ collection and found these two specimens were shrunken and not transparent enough for study. W. J. Wrenn argued that it was unlikely that the two improperly mounted specimens belonged to the common *Eutrombicula* species in the Eastern US, where there is a large number of species in the genus *Eutrombicula* in the US [46,47]. Another synonym on Fuller’s list for this species is *Trombicula cinnabaris*, Ewing, 1920 [49]. D. A. Crossley rechecked Ewing’s specimens named *Trombicula cinnabaris* and found several morphological characteristics were in accordance with the illustrations of *E. alfreddugesi* [46]. Thus W. J. Wrenn concluded that *E. cinnabaris* was the most suitable name for the species that was formerly classified as *E. alfreddugesi* [46,47]. Wrenn continued to use *E. cinnabaris* for his publications [50,51]. In the world checklist [5] and the North American checklist [17] of Trombiculidae and Leeuwenhoekiidae, *E. alfreddugesi* and *E. cinnabaris* were listed as two species. The possible reason for this could be the new name for this species that was proposed, but others treated it as another species and still used the old name. It was suggested by Loomis and Wrenn [47] that the genus *Eutrombicula* needed revision, and this revision is greatly overdue.

### 3.2. Identification

The taxonomy of chiggers almost exclusively relies on larval morphological characteristics [52,53,54,55]. A chigger’s body is divided into two sections, the gnathosoma and idiosoma. The gnathosoma is the chigger mouthparts comprised of four parts: the cheliceral blades, palps, cheliceral base and gnathobase [36]. The shape of the scutum (usually the only sclerotized plate on the idiosoma, next to the gnathosoma), the number and arrangement of dorsal setae, chaetotaxy of legs, leg segmentation, and different forms or length of the gnathosoma (the cheliceral blades, the galeal setae, the palpal setae and the palpal claw), etc. are used for chigger identification.

Chiggers typically were cleared in lactophenol for them to be transparent and easy to observe under the microscope for classification [56,57,58]. After clearing, chiggers are then mounted dorsal-ventrally in mounting media like Hoyer’s medium [59] or Berlese fluid [60] on glass slides and dried at 50 °C before identification under a high magnification microscope using differential interference contrast [61]. More recently methods to clear and mount ectoparasites has improved, e.g., lactic acid, phenol and iodine and water are used for the preparation of rapid clearing solution for chiggers [62].

The identification relies on published dichotomous keys based on morphological characteristics. Some of the published keys are specific for a particular region. For example, Nadchatram and Dohany [63] provided keys for Southeast Asian chiggers to the subfamily, genus and subgenus; Brennan and Goff [64] provided a key to the western hemisphere (Nearctic and Neotropical regions) to the genus level; and Fernandes and Kulkarni [35] provided a key of Indian chiggers to the subfamily, tribe, genus and species. There is also separate guidance for species-level identification. For example, Vercammen-Grandjean and Langston [65] and Stekolnikov [66] provided keys to *Leptotrombidium* which can be used to identify *Leptotrombidium* chiggers into species.

Identifying chiggers using morphological characteristics is complicated and also time-consuming as chiggers are tiny and require days for sample preparation before identifying them under a high magnification microscope. Another challenge is to find a suitable key to guide the identification process, and sometimes no keys are available for certain species since the keys are not up to date. This identification largely relies on personal experiences. Researchers often need to send samples to experts to verify their identification [67,68]. Current knowledge of intraspecific variance is not sufficient, and some species are described from only a few specimens [46]. For measurements of body parts of chiggers, different researchers will likely have different results depending on how the measurements were made. In the future, chigger identification should consider not only morphological characteristics but also molecular tools to improve accuracy.

The molecular data for chigger identification are limited, especially in the US. The nucleotide data on chiggers in GenBank (Access date: 23 March 2022) was examined and summarized in Table 1. There were 10,465 entries in total and molecular data for 47 chigger species. Data for 36 species were published in a peer-reviewed journal while 11 species have not been published. Given the number of species of chiggers worldwide, the molecular data in GenBank is greatly limited, only making up 1.5% (47/3013) of the total number of species. This was surprising considering the significant health risks to humans from these mites. We categorized the published chigger data available in GenBank by country. The molecular data were mostly found in Southeast Asian and South American countries. The only species with molecular data available from the US was for *E. splendens* [69,70] with a complete sequence provided of the small and large subunit ribosomal RNA gene.

We constructed a Neighbor-Joining tree of the small subunit rRNA sequences for chiggers collected from American countries (Figure 5). The chigger small subunit rRNA sequences data were downloaded from GenBank only when the data were also published in peer-reviewed journals. *Eutrombicula tinami*, *Quadraseta trapezoides* and *Trombewingia bakeri* representing different genera, were clustered together in our analysis (Figure 5). *Herpetacarus hertigi*, *Blankaartia sinnamaryi*, and *E. daemoni* are more closely related to each other than they were *E. splendens* and *E. goeldi* in the tree. These results suggest that using the current available sequences in GenBank to identify chiggers would most likely lead to incorrect identifications.

Obtaining chigger molecular data for the US is long overdue. One of the reasons this is lacking is the difficulty in making a morphological identification and at the same time obtaining molecular data. The sample preparation process for chiggers requires the use of lactophenol or acidic phenol for clearing and heat drying (discussed earlier). This process destroys the DNA. There are two solutions available for researchers to resolve this issue. Lee et al. [84] modified the method from Dohany et al. [85] and Ree et al. [86] for the detection of *O. tsutsugamushi* from chiggers using a fluorescent antibody test. Lee et al. [84] used a fine needle to puncture the dorsal-posterior part of the chigger and squeezed out internal tissue for DNA extraction while the exoskeleton was retained mostly intact for morphological identification. Kumlert et al. [71] argued that this method affects the morphological identification, and the DNA extraction process is challenging because of the limited amount of tissue obtained by this method. Kumlert et al. [71] alternatively used the autofluorescent characteristics of the chigger exoskeleton and developed an identification method using a combination of autofluorescent and brightfield microscopy imaging. Chiggers were mounted in sterile water or ethanol to prevent DNA damaged by clearing agents and saved for molecular identification later.

Having a database containing the sequences of mitochondrial cytochrome oxidase subunit I gene (*CO*I) and nuclear genome fragments at a minimum is essential to building a reliable molecular identification method for chiggers. A substantial systematics investment is needed to achieve this objective for chiggers in the US. There is also an increasing interest of using high resolution, three channel imaging coupled with artificial intelligence using multi-layered neural networks to classify closely related species of almost identical-looking moth eggs (Roe, in review) that could be applied to chigger classification. This approach is especially useful for classifying small objects and because the system is automated, has the potential for high throughput.

Chiggers are a poorly understood group of arthropods with a significant importance to human health that *in toto*, are understudied at many levels including systematics and classification. Molecular tools not only could be used to identify the chiggers to species, but also be applicable for classification of the free-living deutonymph, adults and eggs.

## 4. Chigger Microbiome

Ponnusamy et al. [87] extracted DNA from *Orientia tsutsugamushi* infected and uninfected *Leptotrombidium imphalum* laboratory colonies at different life stages: larvae (chiggers), deutonymph, male adults and female adults from Thailand. The 16S rRNA V3-V4 region amplicons were amplified from the extracted DNA and sequenced using Illumina MiSeq. They reported *Orientia*, *Ralstonia* and *Propionibacterium* were the only three taxa that represented >5% of the relative bacteria abundance for *O. tsutsugamushi* infected chiggers. For *O. tsutsugamushi* uninfected chiggers, the *Ralstonia*, *Propionibacterium*, *Streptococcus* and *Corynebacterium* genera represented >5% the bacteria relative abundance [87]. Notably, unidentified taxa of Amoebophilaceae were found predominantly in *O. tsutsugamushi* infected female *L. imphalum* mites but with low relative abundance in larvae (chiggers) and male adult mites. This suggested a mutualistic relationship between Amoebophilaceae and *O. tsutsugamushi* in *L. imphalum* mites that needs further study [87].

Alghamdi [68] extracted DNA from ten species of chiggers collected from rodents in southwestern Saudi Arabia: *Ascoschoengastia browni*, *Ericotrombidium caucasicum*, *Ericotrombidium kazeruni*, *Microtrombicula microscuta*, *Pentidionis agamae Schoutedenichia saudi*, *Gahrliepia lawrencei*, *Schoutedenichia thracica*, *Schoutedenichia zarudnyi* and *Helenicula lukshumiae*. Chigger species were identified using an autofluorescence method [71]. Alghamdi amplified the 16S rRNA V3-V4 region from the extracted DNA and sequenced the 16S rRNA libraries using Illumina. *Corynebacterium*, *Mycobacterium*, *Staphylococcus*, *Candidatus Cardinium*, *Burkholderiaceae* and *Wolbachia* were found to be dominant in all the chigger samples. Especially, *Wolbachia* had a high relative abundance in *P. agamae*. Alghamdi [68] reported that *Orientia* OTUs were observed from *E. kazeruni* and *P. agamae* with a low relative abundance and had 98.88% similarity with the *Orientia chuto* sequence in NCBI. Alghamdi suggested that scrub typhus might exist in Saudi Arabia. More work is needed to confirm the *Orientia* sequence in chiggers.

Chaisiri et al. [88] studied the microbiome of nine chigger species that were collected on small mammals from 11 provinces in Thailand: *Ascoshoengastia indica*, *Blankaartia acuscutellaris*, *Helenicula kohlsi*, *Helenicula pilosa*, *Leptotrombidium deliense*, *Schoengastiella ligula*, *Walchia micropelta*, *Walchia minuscuta*, and *Walchia kritochaeta*. The chiggers were identified by autoflorescent imaging [71]. The 16S rRNA gene V4 region was amplified by PCR and sequenced by Illumina MiSeq. They reported that the dominant taxa across all the chigger samples were *Sphingobium*, *Mycobacterium*, Neisseriaceae and Bacillales. *Geobacillus* was found to be a potential arthropod endosymbiont. *Cardinium, Pseudonocardia and Rickettsiella* were presented in 20–45% of the pooled chigger samples while *Wolbachia* was detected in 3.18% of the pooled samples. *O. tsutsugamushi* and *Borrelia* spp. were detected in *L. deliense*. Potential human pathogens, *Staphylococcus* and *Haemophilus parainfluenzae*, were found in several chigger species.

The microbiome of chiggers from a Thailand lab colony and field samples from Thailand and Saudi Arabia was highly diverse. Not surprisingly, *O. tsutsugamushi* was detected in the Thailand chiggers. However, the finding of *Borrelia* spp. in chiggers [88] suggested they could be the vector for Borreliosis (Lyme disease), currently believed to be transmitted only by ticks. Chiggers might also harbor other human and animal pathogens like *Staphylococcus*. More work is needed to characterize the microbiome of different chigger species from different habitats to better understand if bacteria will affect the vector competence of chiggers, and to better understand if there are any other pathogens that chiggers can vector other than *Orientia.*

## 5. Do Chiggers Transmit *Rickettsia* and *Orientia* in the US?

### 5.1. Summary of Zoonotic Bacteria and Viruses Transmitted by Chiggers

*Leptotrombidium* spp. are vectors of *Orientia tsutsugamushi* that cause Scrub typhus, a life-threatening disease with up to a 50% fatality rate if left untreated [89]. There are 46 chigger species reported to be infected with *O. tsutsugamushi* [90]. Eleven *Leptotrombidium* species were proven to be vectors of scrub typhus [12]. *Leptotrombidium* spp. are both the vector and reservoir of *O. tsutsugamushi* [91]. The high transovarial and transstadial transmission rates among chigger species play an important role in maintaining the disease in nature, since chiggers only feed on an animal host once [92]. Whether rodents also serve as the reservoir for *O. tsutsugamushi* needs further investigation; rodents could only be dead-end hosts [93].

Scrub typhus was originally endemic in the tsutsugamushi triangle. This traditional triangle area extends north to Japan and Russia, south to Northern Australia, and west to Afghanistan and Pakistan [94]. Recent studies suggest scrub typhus is not limited to this triangle. Scrub typhus cases were reported in the United Arab Emirates, and a new *Orientia* species *O. chuto* was isolated from a patient in 2006 [95]. Scrub typhus cases reported in Chile [96,97] and serologic evidence in Peru [98] suggested scrub typhus is endemic in South America. Furthermore, *Orientia* sequences were detected in rodents from France [99], Senegal [99] and Kenya [100]. Besides some *Leptotrombidium* species, *Schoengastiella ligula* has been implicated as a vector of scrub typhus in India [101]. Thus, a larger geographic distribution of scrub typhus, potential new *Orientia* species and new vector chigger species should be taken into account. The potential of *Orientia* species establishing in the US is high given the number of chigger species found throughout the US but has never been seriously studied [17].

Chiggers are also reported to play a role in Hantavirus transmission, a disease previously thought to be transmitted through aerosol exposure to rodent saliva, urine and feces (CDC https://www.cdc.gov/hantavirus/, accessed on 19 May 2022). Hemorrhagic fever with renal syndrome (HFRS) and hantavirus pulmonary syndrome (HPS) are the two clinical syndromes caused by hantavirus when transmitted from rodents to humans [102]. In the United States, hantavirus cause hantavirus pulmonary syndrome (HPS) instead of HFRS, and HPS is a severe and sometimes life-threatening respiratory disease [103]. Wu et al. [104] studied the role of *L. scutellare* in the transmission of hantavirus that cause HFRS in China. They reported hantavirus were isolated from (*i*) rats trapped in the HFRS endemic areas of Shaanxi Province, China and also from chiggers on the rats; (*ii*) from eight previously hantavirus negative mice placed in metal cages in the field to lure chiggers; (*iii*) from mice bitten by virus-positive chiggers; and (*iv*) from larvae hatched from eggs collected from endemic areas. Wu et al. [104] concluded that *L. scutellare* could be the vector of hantavirus and able to transmit the virus to its offspring and rodent hosts by biting. A review of the recent studies of mites and hantavirus in China also indicated that the hantavirus can be maintained in chiggers by transovarial transmission [105]. Rodent ectoparasite screening in Texas detected two chiggers collected from these rodents and one free-living chigger collected from the soil that were positive for hantavirus specific RNA [106]. Although cumulative cases in the US have occurred predominately in the western states, only 10 states have been immune to its presence (https://www.cdc.gov/hantavirus/surveillance/index.html, accessed on 19 May 2022). The possible role of chiggers in the transmission of hantavirus in the US needs further investigation.

*Bartonella tamiae* was isolated for the first time from patients to determine the etiology of three febrile cases in Thailand [107]. In another study, chiggers in the genus *Leptotrombidium*, *Schoengastia*, and *Blankarrtia* were collected from rodents in Thailand, and the 16S–23S intergenic spacer region and citrate synthase gene (*gltA*) region were targeted by PCR to screen for *B. tamiae* DNA in these chiggers [108]. In this work, 29 of the 40 chigger samples tested positive for *B. tamiae* DNA, and there were 97.7–100% sequence homology between the *Bartonella* DNA detected in this study and the *Bartonella* species isolated from three patients in Thailand. This suggested that chiggers play a potential role in the transmission of *Bartonella* to humans.

*Rickettsia* species have also been found in chiggers. *Rickettsia* spp.TwKM02 and TwKM03 were detected in *Leptotrombidium* chiggers from rodents in two islets in Taiwan; the two *Rickettsia* species were close to *R. australis* and *R. felis* URRWXCal2 based on a phylogenetic tree constructed on partial *gltA* gene sequences [109]. *R. australis*, *R. japonica*, *R. felis*, *R. typhi*, *R. akari*, *Rickettsia* sp. TwKM02, *R. conorii* and *Rickettsia* sp. Cf15 were detected in chiggers collected from rodents in South Korea [110]. A new *Rickettsia* species found in *L. scutellare* from rodents in Shandong, China was close to *R. australis* and *R. akari* [111]. *Rickettsia* spp. were detected in chiggers, *Herpetacarus hertigi*, *Quadraseta trapezoides* and *Trombewingia bakeri*, collected from rodents in Brazil and was identical to “*Candidatus* Rickettsia colombianensi” based on the amplified *gltA* and *ompA* genes [79]. *Rickettsia* spp. found from soil-inhabiting unfed *L. scutellare* chiggers in Japan were close to *R. akari*, *R. aeschlimannii, R. felis* and *R. australis* [112]. Since chiggers in this study were unfed, this suggested *Rickettsia* spp. were transstadially and/or transovarially transmitted in chigger mites [112].

Except for *O. tsutsugamushi* that cause scrub typhus, chiggers mostly are not considered as vectors for rickettsioses. Given that some *Rickettsia* spp. found in chiggers were similar to species in the spotted fever group (SFG), e.g., *R. aeschlimannii*, *R. felis* and *R. australis* (recognized because of their infectivity and pathogenicity as *Rickettsia* spp.), the possible role chiggers might play in the transmission of *Rickettsia* to humans deserves further investigation. A well, a pool of chiggers with *Rickettsia* spp. TwKm02 were positive for *O. tsutsugamushi* in Taiwan [109]. Therefore, the possibility of coinfection of SFGR and scrub typhus in chiggers needs further study. Additionally, *R. felis* and other *Rickettsia species* were detected in chiggers collected from rodent in the US [23]. It is possible that rickettsioses in the US attributed to ticks might also originate from chiggers. Since the chigger microbiomes in the US have been minimally studied, this remains a possibility.

*Borrelia* species have also been found in chiggers. *Borrelia burgdorferi* sensu lato (s.l.) was found in 634 *Neotrombicula autumnalis* chiggers collected on 100 wild-caught rodents along with 1380 questing chiggers on vegetation, using PCR and DNA hybridization in Germany and *Borrelia* DNA was detected from a chigger collected from a greater white-tooted shrew [113]. An infection experiment also was conducted by Kampen et al. [113] by placing 305 questing unfed chiggers on *B. garinii* infected laboratory mice and on *B. afzelii* infected laboratory Mongolian gerbils. *Borrelia* DNA was detected on a pool of four chiggers that fed on *B. garinii* infected laboratory mice, and a nymph kept from a larva that was previously fed on *B. afzelii* infected laboratory Mongolian gerbils. Results suggested a possible transmission of *Borrelia* spp. from *N. autumnalis* to mice and gerbils. *Borrelia* spp. DNA was also screened on 509 chiggers in the genus *Neotrombicula* that were collected from 1080 wild birds in Certak, Czech Republic [114]. One sample pooled with five *Neotrombicula* chiggers removed from one Eurasian blackcap (*Sylvia atricapilla*) tested positive for *B. burgdorferi* s.l. by PCR and a reverse line blotting assay [114].

### 5.2. Role of Endosymbionts in Orientia Infection

Takahashi et al. [115] reported that the adults in *O. tsutsugamushi* infected *Leptotrombidium fletcheri*, *L. arenicola* and *L. deliense* colonies were almost entirely made up of females, and this female-biased sex ratio represents a feminization. After treating the *O. tsutsugamushi* infected *L. fletcheri* colonies with antibiotics, a high ratio of adult males occurred. Takahashi et al. [115] suggested *O. tsutsugamushi* inhibits the development of males. Recent microbiome studies detected *Wolbachia* and *Cardinum* sequences in chiggers [68,88]. *Wolbachia* and *Cardinum* are facultative bacterial endosymbionts of arthropods, commonly maternally inherited and associated with reproductive manipulation such as feminization and parthenogenesis [116]. It is possible that *Wolbachia* and *Cardinum* can also cause feminization in some chigger species infected with *O. tsutsugamushi.*

Feminization also occurred in *O. tsutsugamushi* infected F1 generation adults of *L. imphalum* [117]. A microbiome study of a *L. imphalum* laboratory colony found an unidentified species, Amoebophilaceae, which dominated in *O. tsutsugamushi* infected female adults and was at a very low relative abundance in *O. tsutsugamushi* uninfected mites [87]. The results suggested a mutualistic relationship between *O. tsutsugamushi* and the unidentified species in the Amoebophilaceae [87]. It is possible that this unidentified species in Amoebophilaceae was responsible for the feminization of *L. imphalum* adults. Further characterization and pathogenicity of this Amoebophilaceae species is needed including understanding their possible contributing role in pathogenesis with other bacteria.

## 6. Possible Misdiagnosis of Tick-Borne Diseases and Allergies

Tick-borne diseases in the US include spotted fever rickettsioses, Lyme disease, ehrlichiosis, anaplasmosis, tularemia, babesiosis, Colorado tick fever, and relapsing fever [118]. Until recently, most patients with tick-associated rickettsiosis were considered to have Rocky Mountain spotted fever (RMSF), caused by *R. rickettsii*. Serological techniques like indirect the immunofluorescence assay (IFA) was commonly used for the routine diagnosis of RMSF [119,120]. However, other *rickettsia* species in the Spotted Fever Group (SFG) can also elicit antibodies that are cross-reactive with *R. rickettsia* (serological cross-reactions) [118]. Additionally, other SFG rickettsia species, including *R. parkeri* and *Rickettsia* species 364D [121] can cause illness that was diagnosed as RMSF. SFG *rickettsia* species have been detected in chiggers [79,109,110,111,112,122], suggesting that reported tick-associated rickettsiosis may have some level of chigger associated.

*Borrelia* spp. DNA was detected from *Neotrombicula autumnalis* collected from wild birds in the Czech Republic [114]. Additionally, a recent microbiome study of *L. deliense* collected from rodents in Thailand showed that *Borrelia* spp. were found in 49.2% of the chigger samples [88]. Larval ticks, sometimes known as “seed ticks,” bite humans and are nearly impossible for non-professionals to distinguish from chigger bites. In a sero-epidemiological study conducted on France forestry workers, a large number of people who were seropositive to *B. burgdorferi* were unable to recall a tick bite [123]. The reasons could be those workers were bitten by larval ticks or chiggers.

It is now well recognized that the carbohydrate galactose-α-1,3-galactose (alpha-gal) which is present in some tick species can elicit an alpha-gal specific IgE immune response [124]. People who were reported to be bitten by ticks have developed a hypersensitivity to red meats, including beef and lamb, called red meat allergy [125]. The symptoms of red meat allergy range from mild to severe, some can be acute and fatal [124]. Notably, people in three cases who had severe red meat allergy later reported having pruritic bites from chiggers either on their lower legs or at the waistband area [126]. It is possible that chiggers could also contribute to the prevalence of red meat allergy. The searching for alpha-gal antigens in chiggers should be investigated to determine if chiggers can also cause red meat allergy.

## 7. Chigger Control

Chiggers are frequently difficult to control chemically [127]. For successful management, large areas would have to be sprayed numerous times [127]. The best way to prevent chigger bites is to avoid them [128]. The methods of avoiding chigger bites are summarized in Table 2.

When venturing into chigger habitat, wearing permethrin-treated clothing and applying repellents containing DEET (diethyltoluamide) is effective against chiggers [129]. Wearing tightly woven long sleeves, long pants, closed-toe shoes and tucking shirts into pants and tucking pants into socks can keep chiggers on the outside of your clothing. There are also non-insecticidal garments available commercially like Rynoskin Total that claim to protect from chiggers, mainly based on preventing access under the garments.

When outdoors, stay in the middle of trails and avoid high bushes and tall grass that might have chiggers. It is also suggested to avoid sitting directly on the ground to reduce chigger encounters [130]. Upon return, it is recommended to remove clothing and wash clothing immediately. It is also helpful to wipe down shoes and boots to avoid bringing chiggers into the home. Taking a hot and soapy shower with a vigorous skin massage will help remove chiggers attached to your body [128].

To reduce chigger populations in the yard, keep turf grass short and vegetation trimmed, since high temperature and low humidity conditions are inhospitable for chiggers. Another approach is to keep small mammals out of the yard using fencing and keeping trash can lids secured to make the trash less attractive [129].

## 8. Conclusions

Overall, chiggers have been understudied world-wide but especially so for the US. The published research mainly focuses on taxonomy (identification methods), ecology, and reports of the discovery of chiggers on various hosts in the US. However, the life history of other stages other than larvae needs further study.

There are 248 chigger species reported from the U.S. to date. Chiggers are reported in every state except Hawaii in the United States. The taxonomy of chiggers almost exclusively relies on larval morphological characteristics; molecular data on chiggers are limited. It is likely that the most common chigger in North America, *E. alfreddugesi*, has been misidentified and should be replaced with *E. cinnabaris*. Building a reliable molecular identification method for chiggers correlated with morphological explicit characters is greatly needed. The molecular data are needed to better identify and study not only the parasitic stage but also free-living deutonymphs and adults correlated to what we already know about the larval stage. Three microbiome studies of chiggers were conducted from Thailand *L. imphalum* lab colonies, ten chigger species collected from rodents in Saudi Arabia, and nine chigger species collected from small mammals in Thailand. More work is needed in this area to characterize the microbiome of different chigger species from different habitats to better understand if bacteria will affect the vector competence of chiggers, and to better understand if pathogens in addition to *Orientia* are transmitted by these mites. Thus far, data suggests pathogen competence in chiggers for scrub typhus might be associated with co-infection with other bacteria.

Scrub typhus was reported outside of the Tsutsugamushi Triangle. In addition to *Leptotrombidium* species, new chigger species were reported to vector the bacteria that causes scrub typhus. The potential of *Orientia* species establishing in the US is high given the number of species found throughout the US but has been minimally investigated. Chiggers are reported to harbor Hantavirus, *Bartonella*, *Rickettsia* and *Borrelia* suggesting a need for further study in North America. The role that chiggers play in disease transmission dynamics in the US is essentially unknown. It is possible some of the tick-borne diseases and red meat allergies are caused by chiggers.

## Figures and Tables

**Figure 1 ijerph-19-15147-f001:**
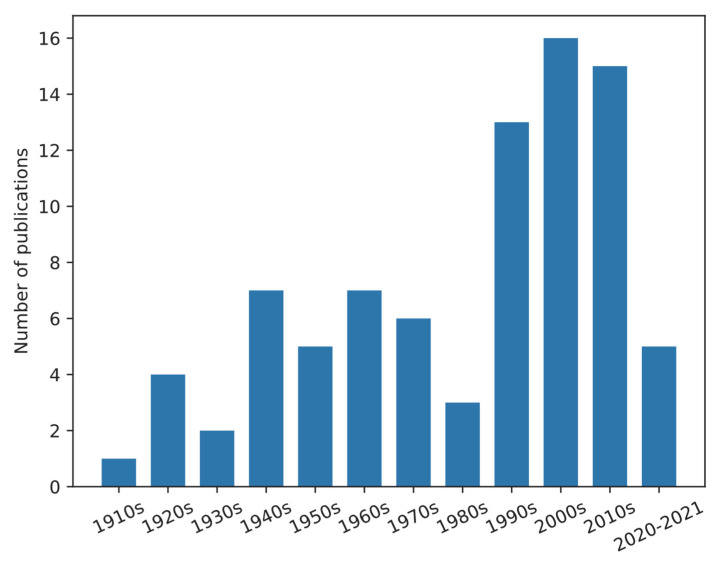
Bar chart of publications of chiggers in the US. Publications were searched through the Web of Science on 3 May 2022. The search strategy is in Appendix A).

**Figure 2 ijerph-19-15147-f002:**
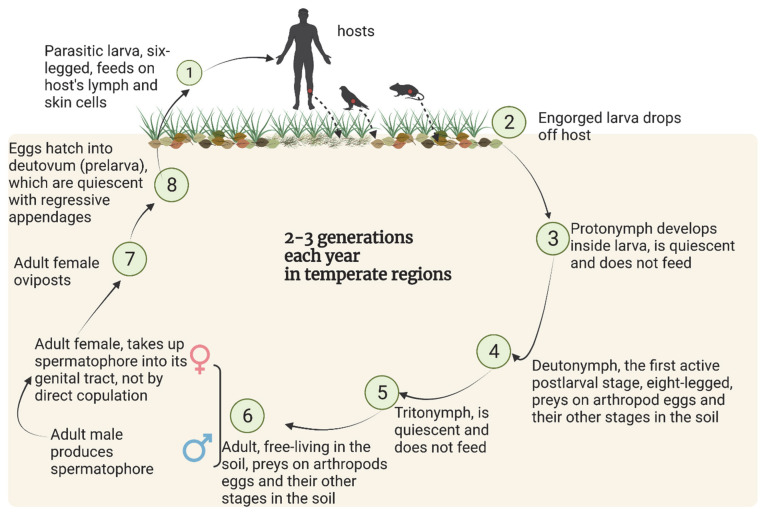
The typical life cycle of chiggers. Created with BioRender.com. Data sources: [11,13,14].

**Figure 3 ijerph-19-15147-f003:**
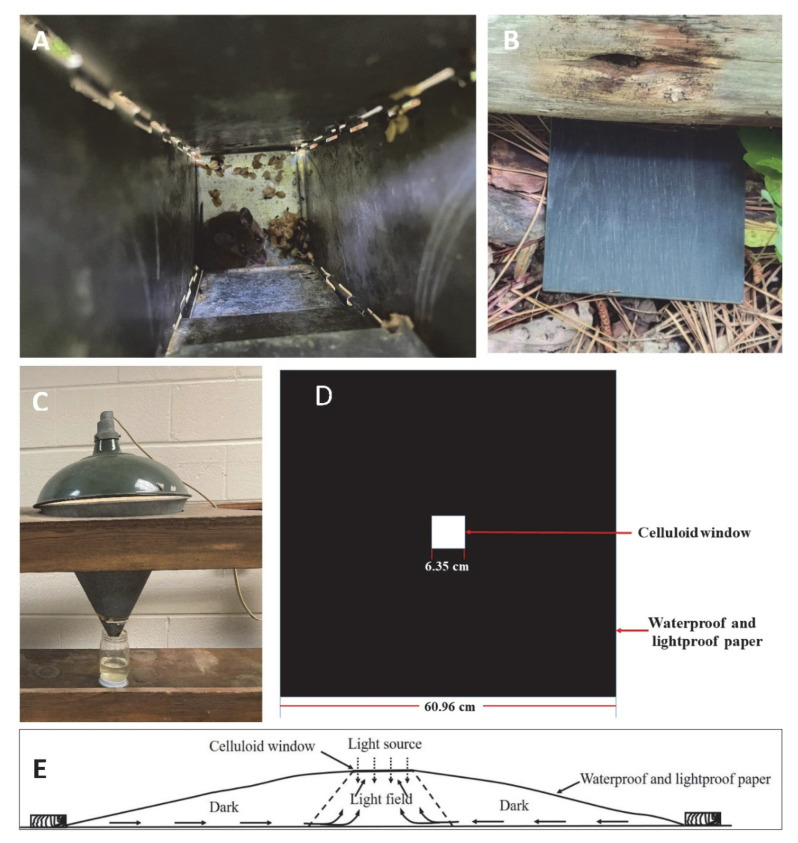
Chigger collecting methods: (**A**) live-trapped rodents in Sherman traps (photo credit to Reuben Garshong); (**B**) black tile on the ground; (**C**) Berlese funnel (photo credit to Steve Denning); (**D**) light trap (modified from [29]); (**E**) horizontal view of light trap, arrows inside the trap show the paths taken by chiggers (modified from [29]).

**Figure 4 ijerph-19-15147-f004:**
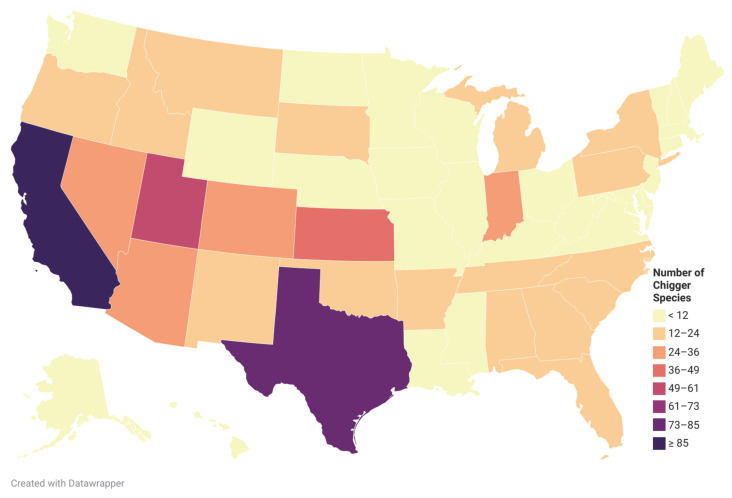
Chigger species richness in the US. Created with Datawrapper. Data sources: [17,45].

**Figure 5 ijerph-19-15147-f005:**
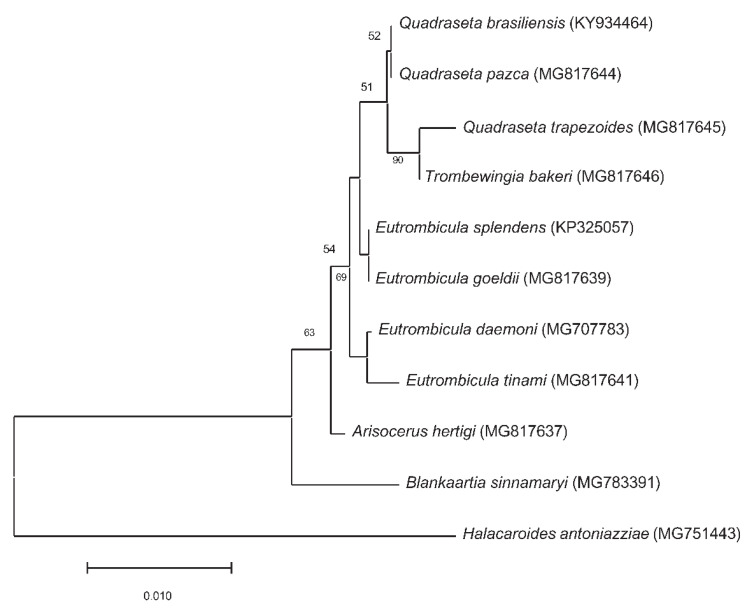
Neighbor-Joining tree of small subunit rRNA sequences of chiggers collected from North and South American countries. The chigger small subunit rRNA sequences data were downloaded from GenBank only from peer-reviewed journal publications. The *Halacaroides antoniazziae* (MG751443) was used as an outgroup. The percentage of replicate trees in which the associated taxa clustered together in the bootstrap test (1000 replicates) are shown below the branches. The tree was drawn to scale, with branch lengths in the same units as those of the evolutionary distances used to infer the phylogenetic tree. The evolutionary distances were computed using the Kimura 2-parameter method and are in the units of the number of base substitutions per site. This analysis involved 10 nucleotide sequences. All positions containing gaps and missing data were eliminated (complete deletion option). There were a total of 403 positions in the final dataset. Evolutionary analyses were conducted in MEGA11.

**Table 1 ijerph-19-15147-t001:** Summary of chigger molecular data in GenBank in peer-reviewed publications.

Country (Continent)	Different Chigger Species in GenBank	% *	Peer Reviewed Publications	Chigger Species Reported	Target Genes *
Laos (Asia)	11	30.56	[71]	*Ascoschoengastia indica*	COI
				*Blankaartia acuscutellaris*	COI
				*Leptotrombidium deliense*	18S, COI, whole genome shotgun
				*Microtrombicula chamlongi*	COI
				*Schoengastia kanhaensis*	COI
				*Walchia alpestris*	COI
				*Walchia ewingi ewingi*	COI
				*Walchia ewingi lupella*	COI
				*Walchia kritochaeta*	COI
				*Walchia micropelta*	COI
				*Walchia minuscuta*	COI
Thailand (Asia)	1	2.78	[72]	*Leptotrombidium deliense*	Mitochondrial (mt) DNA complete genome
China (Asia)	1	2.78	[73]	*Neoschoengastia gallinarum*	small subunit rRNA, large subunit rRNA, 5.8S, internal transcribed spacer 2, large subunit rRNA, COI
Korea (Asia)	2	5.56	[74]	*Helenicula miyagawai*	18S
				*Leptotrombidium scutellare*	mt DNA complete genome, COI, 18S, 5.8S
Japan (Asia)	1	2.78	[75]	*Leptotrombidium pallidum*	COI, mt DNA complete genome, 5.8S, whole genome shotgun
Malaysia (Asia)	1	2.78	[76]	*Leptotrombidium fletcheri*	mt DNA complete genome
Canada (North America)	1	2.78	[77]	*Neotrombicula microti*	COI
US (North America)	1	2.78	[69,70]	*Eutrombicula splendens*	large subunit rRNA, small subunit rRNA, 28S, 18S, *Ef1-α*, *Srp54*, *Hsp70*, COI
Mexico (North America)	1	2.78	[70]	*Acomatacarus arizonensis*	28S, 18S, COI, *Ef1-α*, *Srp54*, *Hsp70*
Brazil (South America)	12	32.33	[78,79]	*Herpetacarus*(*Arisocerus*) *hertigi*	small subunit rRNA
				*Blankaartia sinnamaryi*	small subunit rRNA
				*Eutrombicula daemoni*	small subunit rRNA
				*Eutrombicula goeldii*	small subunit rRNA
				*Eutrombicula tinami*	small subunit rRNA
				*Fonsecia ewingi*	18S
				*Hannemania hepatica*	18S
				*Hannemania yungicola*	18S
				*Quadraseta brasiliensis*	small subunit rRNA, 18S
				*Quadraseta pazca*	small subunit rRNA
				*Quadraseta trapezoides*	small subunit rRNA
				*Trombewingia bakeri*	small subunit rRNA
Poland (Europe)	3	8.33	[80,81]	*Hirsutiella zachvatkini*	COI
				*Miyatrombicula muris*	COI
				*Neotrombicula inopinata*	COI
Spain (Europe)	1	2.78	[82]	*Morelacarus* sp.	COI
Madagascar (Africa)	1	2.78	[83]	*Schoutedenichia microcebi*	small subunit rRNA

* %: Percent of different chigger species from each country out of the total 36 different species in GenBank in peer-reviewed publications. * Gene abbreviations: COI, cytochrome *c* oxidase I; 18S, 18S ribosomal RNA gene; 5.8S, 5.8S ribosomal RNA gene; *Ef1-α*, elongation factor 1alpha100E; *Srp54*, signal recognition particle protein 54k gene; *Hsp70*, Hsc70-5 heat shock protein cognate 5 gene.

**Table 2 ijerph-19-15147-t002:** Methods of avoiding chigger [127,128,129].

Scenarios	Methods of Avoiding Chigger Bites
When venturing into chigger habitat	Wear permethrin-treated clothing
	Apply repellents like DEET
	Wear tightly woven long sleeves, long pants, and closed-toe shoes
	Tuck shirts into pants, tuck pants into socks
	Stay in the middle of the trail, avoid high bushes and tall grass
	Avoid sitting directly on the ground
After you come inside	Remove clothing immediately and launder it in hot water
	Wipe down shoes and boots
	Take a hot and soapy shower with a vigorous skin massage
Yard	Try to keep small mammals out of the yard by putting up a fence and keeping trash can lids secured
	keep turf short and vegetation trimmed

## Data Availability

Not applicable.

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
