# Peer review of "Biology, Systematics, Microbiome, Pathogen Transmission and Control of Chiggers (Acari: Trombiculidae, Leeuwenhoekiidae) with Emphasis on the United States"

_ijerph, 2022, doi:10.3390/ijerph192215147_

Round 1
Reviewer 1 Report (Previous Reviewer 1)
I am satisfied with the authors' reply to my notes and with the corrections following my suggestions. Attached is the manuscript with some minor corrections.

Author Response
Reviewer Comments:
I am satisfied with the authors' reply to my notes and with the corrections following my suggestions. Attached is the manuscript with some minor corrections.
Authors’ response: We thank this reviewer for comments and suggestions for change. We have revised the submitted manuscript per the suggestions indicated on the PDF copy of the manuscript. The corrections were made in track changes for your consideration.
Reviewer 2 Report (Previous Reviewer 2)
The present study has some points that need to be reanalyzed and reevaluated to be resubmitted for a new review. As it is presented, there is no way to accept it for publication. Reading the text, I don't see any importance or any innovation in the present study, what is the importance of the work¿ What has been added? And yet, before the text is corrected, the main points listed below need to be considered:
(1) The title does not match the work. At first glance, reading the title, I thought that innovative and unpublished information would come, but throughout the text, they were only compilations from other places, without any more information, and they brought several basic taxonomic errors.
(2) The authors wrote: “In addition, several other recognized pathogens to infect humans, namely Hantavirus, Bartonella, Borrelia, and Rickettsia, were also identified in chiggers.” I believe the term is used incorrectly here. The possible pathogens were not identified but detected through molecular techniques. Which is very different information from what was said in the abstract, and throughout the text.
(3) Figure 3 is not formatted, each figure out of the standard of a properly suited Figure board.
(4) About “Chigger Lab Rearing”. There is current information on the creation of chiggers in the laboratory that were not even mentioned in the present text. Showing lack of expertise of the authors.
(5) The suggestion given in the '3.1 Taxonomy' section about the revision of Eutrombicula is necessary, it is not an innovative idea of the present work. Several current works suggest the same. please credit the people who have already suggested the same as they are suggesting.
(6) Arisocerus hertigi - apparently the authors are not taxonomists of the group. This species is no longer in that genus, as well, this genus no longer exists.
(6) The present authors wrote: “These results suggest that using the current available sequences in GenBank to identify chiggers would most likely lead to misidentifications”.
The above statement is used deliberately and irrationally, since 18S is not useful for species identification because it is a very conserved gene, the single tree presented does not have an outgroup, much less has been rooted. Also, the bootstrap values are quite low and no discussion of this has been made. Therefore, claiming that the information published on the genbank leads to the wrong identification of mites, at the very least, is wrong and is disrespectful. I believe a taxonomist on chiggers needs to be consulted.
In view of this, the present work should not be accepted. The work is nothing more than a compilation of previously published data, without any additional information, all the information placed here has already been written by other authors, and in the present work, instead of making the correct citations, they are said by the authors as if it was the first time. And this is not true. Furthermore, the lack of expertise in chiggers is easily detectable in several parts of the work.
Author Response
Reviewer Comments: The present study has some points that need to be reanalyzed and reevaluated to be resubmitted for a new review. As it is presented, there is no way to accept it for publication. Reading the text, I don't see any importance or any innovation in the present study, what is the importance of the work? What has been added? And yet, before the text is corrected, the main points listed below need to be considered:
Authors’ response: The primary purpose for this mini-review was to report on what we know in the US and do not know in the US (based on studies outside of the US) about chigger research in the US. This paper is a mini review of the literature (the last review was several years ago) from which conclusions were made about the “state of the science of chiggers” in the US and what work is needed in the future in the US.
Reviewer Comments (1): The title does not match the work. At first glance, reading the title, I thought that innovative and unpublished information would come, but throughout the text, they were only compilations from other places, without any more information, and they brought several basic taxonomic errors.
Authors’ response: We do not understand the reason for this comment, since above the title of our paper is the title, REVIEW. We can add review to our submitted title if the editor does not consider this addition redundant. Editor, please provide instructions on this issue. This paper is a review of the literature from which conclusions were made about the “state of the science of chiggers” in the US and what work is needed in the future. For this reason, this paper was submitted to the journal in the category of a “review”.
Taxonomical errors were corrected and specified further below.
Reviewer comment (2): “In addition, several other recognized pathogens to infect humans, namely Hantavirus, Bartonella, Borrelia, and Rickettsia, were also identified in chiggers.” I believe the term is used incorrectly here. The possible pathogens were not identified but detected through molecular techniques. Which is very different information from what was said in the abstract, and throughout the text.
Authors’ response: Thank you. Changes were made as suggested by the reviewer. Please refer to lines 25 and 594.
Reviewer comment (3): Figure 3 is not formatted, each figure out of the standard of a properly suited Figure board.
Authors’ response: We have formatted Figure 3 following the journal's requirements.
Reviewer comment (4): About “Chigger Lab Rearing”. There is current information on the creation of chiggers in the laboratory that were not even mentioned in the present text. Showing lack of expertise of the authors.
Authors’ response: We conducted an additional literature search in October 2022 in response to this Reviewer’s Comment for published research on chigger lab rearing methods in Google scholar, PubMed, Web of Science, and Scopus databases, but no additional peer-reviewed publications were found. We were already aware of a paper published in 1954 for maintaining chiggers collected from the field in the lab until they became nymphs. Since this method was not for establishing a continuous lab colony, we did not include this citation. Another paper published in 1946 about chigger rearing was not cited since this method was outdated and was believed to be not successful for rearing a wide range of species by Nadchatram 1968. We are not sure if those two citations are the reason for the reviewer’s concern. We have now added the two citations to our review in case this is the Reviewer’s concern. We would be happy to include any additional current publications on chigger lab rearing if this reviewer could provide a citation of the article of concern.
Reviewer comment (5): The suggestion given in the '3.1 Taxonomy' section about the revision of Eutrombicula is necessary, it is not an innovative idea of the present work. Several current works suggest the same. please credit the people who have already suggested the same as they are suggesting.
Authors’ response: We did not write that the suggestion of the revision of Eutrombicula was our innovative idea. Actually, we stated in lines 319-321, “Loomis and Wrenn [45] investigated the systematics of the genus Eutrombicula and found that this genus in the Eastern US needed revision since some of the records was misidentified.” Although we cited the Loomis and Wrenn’s paper in this manuscript multiple times, we will also add this information further in lines 338 to 339.
Reviewer comment (6): Arisocerus hertigi – apparently the authors are not taxonomists of the group. This species is no longer in that genus, as well, this genus no longer exists.
Authors’ response: Arisocerus hertigi was listed in Table 1, a summary of chigger molecular data as it appears in GenBank associated with peer-reviewed publications. The aim was to summarize these data as they appear in GenBank. The goal was not to review the history of this genus. Arisocerus hertigi was the name appearing in GenBank (Accession number: MG817637). In this revision of our review, Arisocerus was changed to Herpetacarus but the name is still unchanged in GenBank and out of our control to update.
Reviewer comment (7): The present authors wrote: “These results suggest that using the current available sequences in GenBank to identify chiggers would most likely lead to misidentifications”. The above statement is used deliberately and irrationally, since 18S is not useful for species identification because it is a very conserved gene, the single tree presented does not have an outgroup, much less has been rooted. Also, the bootstrap values are quite low and no discussion of this has been made. Therefore, claiming that the information published on the genbank leads to the wrong identification of mites, at the very least, is wrong and is disrespectful. I believe a taxonomist on chiggers needs to be consulted.
Authors’ response: Our rationale was explicit in the review but appears to not have been considered in the Reviewer’s comment. Maybe, there is a better way to highlight this point or maybe the reviewer did not see this explanation. We don’t understand the reason for this reviewer’s comment because we don’t disagree with the reviewer.
We agree with the reviewer that the 18S sequences are not helpful for species determination. We understand the problems with 18S. This was the point of our analysis of the molecular data. The available US chigger molecular data in toto in peer-reviewed publications with corresponding records in GenBank are limited (see Table 1) with limited information on different gene targets. Therefore, we had only two options available for our analysis relative to understanding what we know about chiggers in the US: 1) construct a tree based on COI data from Laos (which was geographically not the focus of our review) or 2) construct a tree based on the small subunit rRNA gene from North and South America (the geographical emphasis of our review). We chose the latter because there was no other options for the Americas and the focus of our review was what we know and not know about chiggers in the US. Our conclusion was correct and in agreement with the reviewer, that these data are not informative for systematics and sequence data for other gene targets are needed.
A new phylogenetic tree was created with an outgroup. Bootstrap values were improved as suggested. The new phylogenetic tree was added to the revised manuscript but did not change the conclusion of the analysis which was consistent with the reviewers view of these same data.
Reviewer comment (8): In view of this, the present work should not be accepted. The work is nothing more than a compilation of previously published data, without any additional information, all the information placed here has already been written by other authors, and in the present work, instead of making the correct citations, they are said by the authors as if it was the first time. And this is not true. Furthermore, the lack of expertise in chiggers is easily detectable in several parts of the work.
Authors’ response: Our mini-review is the “first ever” review of “what we know and do not know” about the main aspects of chiggers in the US, thus the reason for our title. It was written in a mini-review format and open access to provide rapid availability to the scientific community of the chigger literature with the emphasis on “what we know and do not know” about chiggers in the US. Not only is it the first ever review of chiggers with an emphasis on the US but also the first ever review like this on chiggers that includes microbiomics. The last review on chiggers was published as a peer reviewed book chapter in 2015. The latest review of US chigger research for a single aspect of chigger biology, systematics, was by Walters et al. 2011. So it has been several years since the chigger literature was reviewed. Our paper was submitted as a review to the journal.
Reviewer also commented, “in the present work, instead of making the correct citations, they are said by the authors as if it was the first time.” We regret that the reviewer feels we did not fairly cite papers for the information presented in the review. The general introduction of the paper is 35 lines long in the final journal format. In these 35 lines which includes Table 1, we cited 89 publications, using the method of citation recommended by the journal, i.e., numbers to designate a specific reference in the references cited section. The next section, life cycle, was 35 lines long and we cited 14 references, the next section on Host and Habitat was 23 lines with 12 citations, the next section on collecting cited 18 references, etc. We cited 192 different papers in our mini review. The use of numbers instead of using the authors’ names to cite papers is the method of citations required by this journal.
Reviewer also commented: “Furthermore, the lack of expertise in chiggers is easily detectable in several parts of the work.” This issue was address in detail in earlier responses to this reviewer and was not shared by the other two reviewers of this paper.
Reviewer comment (9): The result is nothing more than a compilation of previously published data, without any additional information, other authors have already written all the information placed here….”
Authors’ response: Our comment to this response was made earlier.
Reviewer 3 Report (New Reviewer)
The authors have presented a comprehensive review on several important aspects of chigger mites, which I think is a good introduction to readers interested in chigger mites and scrub typhus.
Author Response
Reviewer Comments: The authors have presented a comprehensive review on several important aspects of chigger mites, which I think is a good introduction to readers interested in chigger mites and scrub typhus.
Authors’ response: We thank this reviewer for the encouraging comments on our manuscript.
This manuscript is a resubmission of an earlier submission. The following is a list of the peer review reports and author responses from that submission.
Round 1
Reviewer 1 Report
The manuscript represent a useful review focused on a series of aspects connected with the medical importance of chigger mites. It was especially interesting to read such rarely reviewed information, as the data on microbiome of chiggers, including the effect of feminization caused by some infections. I hope that this review will have positive impact on the chigger studies in US, which greatly decreased in last decades. Taking into account that a comprehensive checklist of North American chiggers was published recently (Walters et al. 2011), a relative omission of the taxonomical aspect of chigger studies in the paper under review seems justified.
I suggest the following corrections in the manuscript.
1) Lines 11 and 37. The statement on Leptotrombidium and Herpetacarus as vectors of Orientia tsutsugamushi seems partly erroneous, since Herpetacarus was found to be infected with another species of Orientia.
2) Table 1. I would recommend including references in the form “Ewing & Hartzell, 1918” to the first column, in addition to numeric ones, for a better understanding.
3) The paragraph on the morphological grounds of the chiggers’ identification (lines 302-308) looks weak. The structure of gnathosoma is described incompletely: the gnathobase (fused palpal coxae), which passes into hypostome, is omitted in the text. The dorsal position of the scutum (not “scuta”!) should be mentioned. “The arrangement of dorsal setae” – should be “number and arrangement”. The chaetotaxy of legs, which is one of the main characters serving for identification of chiggers at the genus and species levels, is omitted. I recommend a revision of this paragraph on the base of more thorough consultation with the recent literature on the morphology of chiggers.
4) Line 309. Clearing in lactophenol can be excluded, as the Hoyer’s and Berlese’s mediums provide enough clearing themselves. Moreover, an excessive clearing can be worse than insufficient for the microscope examination.
5) Line 312. “Identification under a high magnification microscope”. I believe that a note on the obligate using of a contrast (preferably, differential interference contrast) should be added.
6) Line 419. “Microtrombicula microscuta sp. nov.”, "Schoutedenichia saudi sp. nov.” The indices “sp. nov.” can only be used in the paper containing original descriptions of these species; please, exclude them.
7) Line 440. A misprint: “delicense” – deliense.
Reviewer 2 Report
The present work has some points that need to be reanalyzed and reevaluated to be resubmitted for a new review. As it is presented, there is no way to accept it for publication. Reading the text, I don't see any importance or innovation in the present study. What is the significance of the work? What has been added? Furthermore, I don't know if this work suits this journal. And yet, before the text is corrected, the main points listed below need to be considered:
(1) The title does not match the work. At first glance, reading the title, I thought that innovative and unpublished information would come. Still, throughout the text, the authors only compilations from other places, without any more information, and they brought several basic taxonomic errors.
(2) The authors wrote: "Chiggers are also reported to be associated with Hantavirus, Bartonella, Rickettsia and Borrelia transmission" For the chigger mites were only confirmed to be transmitters of Orientia, while for the other pathogens, the chiggers were found to be naturally infected, which is very different information from what was said in the abstract, and throughout the text.
(3) The following sentence highlights the group's lack of expertise regarding the taxonomy and systematics of chiggers: "Chiggers in the genus Hannemannia primarily fed on reptiles [73]" – First, Hannemania is misspelled, and secondly, none of Hannemania described until today, was found parasitizing reptiles, this genus is specialized in parasitizing amphibians.
(4) What is the importance of table 1? Most species are not in italics, are not taxonomically correct (several species have already been synonymized), and there is still a lot of information missing and species that were not listed. Publishing a table like this will only cause more confusion to the chigger group.
(5) The authors wrote: "Chiggers were typically cleared in lactophenol ……". According to whom? The vast majority of taxonomists no longer use lactophenol because it is toxic. This solution has long since been replaced.
(6) The present authors wrote: "These results suggest that using the current available sequences in GenBank to identify chiggers would most likely lead to misidentifications".
The above statement is used deliberately and irrationally since 18S is not helpful for species identification because it is a very conserved gene. The single tree presented does not have an outgroup, much less has been rooted. Also, the bootstrap values are pretty low, and no discussion has been made. Therefore, claiming that the information published on the Genbank leads to the wrong identification of mites, at the very least, is inappropriate and disrespectful. I believe a taxonomist on chiggers needs to be consulted.
Given this, the present work should not be accepted. The result is nothing more than a compilation of previously published data, without any additional information, other authors have already written all the information placed here, and in the present work, instead of making the correct citations, they are said by the authors as if it was the first time. And this is not true. Furthermore, the lack of expertise in chiggers is readily detectable in several parts of the work.
Reviewer 3 Report
Dear Editor and Authors,
The manuscript presented attempts to review chigger mites in several biological, ecological, and taxonomic aspects, emphasizing the USA. The review idea is good but poorly executed because the authors are not experts in chiggers. Unfortunately, I do not recommend the publication of this manuscript as several data presented are not robust enough, present holes in knowledge, are outdated, and do not match the proposed title. In addition, several data from other works are presented as results of the present manuscript, and the proper credit was not given to the authors who made them (plagiarism). Below are more details:
1. Introduction. It must be reviewed! When the authors cite data, they should refer to it, as several of the information is not the present study's results. The article search methodology is weak because it only relies on the Web of Science collection, and several citations are missing, not corresponding to the reality of what is known in the USA. Table 1 is extremely extensive, and most of the species presented are outdated, which creates confusion in the correct identification to which the authors are referring. I believe authors should contact a chigger expert to resolve this erroneous data. I recommend looking for the researcher Ricardo Bassini-Silva, who worked with the fauna of the EUA, and I believe he can help with the update.
2) Chigger Biology. There are erroneous data because the host's tissue encompasses species. Figure 2 is based on some cycles. Authors need to include citations or data if they have collected it. Species have different life cycles within families.
3) Hosts and Habitat. The genus Hannemania does not parasite reptiles. In addition, the authors mix data from authors who studied species from different countries.
4) Collecting Methods. The authors state that the same methodology is used to capture different host orders. The authors probably never worked with all these groups, as each requires a diverse collection methodology.
5) Chigger Lab Rearing. This breeding method does not work for some species. Not all of them feed on Collembola. It needs to be reviewed.
6) The Neighbor-Joining tree has errors because few terminals were used, and, in addition, it is already known that the 18S gene is not appropriate to define species.
7) 4. Microbiome Chigger. No USA data was presented.
8) Do Chiggers Transmit Rickettsia and Orientia in the US? The same applies. Studies with pathogen detection have already been carried out in the USA and were not cited. Authors must focus on the objective, have factual data, and avoid speculation.